# Saliva Analysis of pH and Antioxidant Capacity in Adult Obstructive Sleep Apnea Patients

**DOI:** 10.3390/ijerph192013219

**Published:** 2022-10-14

**Authors:** Nicolò Venza, Giulia Alloisio, Magda Gioia, Claudio Liguori, Annarita Nappi, Carlotta Danesi, Giuseppina Laganà

**Affiliations:** 1Department of Systems Medicine, University of Rome ‘Tor Vergata’, Via Montpellier 1, 00133 Rome, Italy; 2Department of Clinical Sciences and Translational Medicine, University of Rome ‘Tor Vergata’, Via Montpellier 1, 00133 Rome, Italy; 3Sleep Medicine Centre, Department of Systems Medicine, University of Rome Tor Vergata, 00133 Rome, Italy; 4Faculty of Medicine, UniCamillus—Saint Camillus International University of Health and Medical Sciences, Via Sant’ Alessandro 8, 00131 Rome, Italy

**Keywords:** OSAS, saliva, pH, antioxidant capacity

## Abstract

Background: Obstructive sleep apnea syndrome (OSAS) may be associated with and activates the stress response system, and variation in the physiological antioxidant capacity of body fluids. Our aim was to evaluate the variation of pH and antioxidant capacity on the saliva of obstructive sleep apnea subjects (OG) compared to a control group (CG). Method: Fifty subjects with moderate/severe OSAS were recruited in Tor Vergata Hospital and compared with 20 healthy subjects CG. The buffer and the antioxidant capacity of the samples were quantified measuring the pH and the percentage of total salivary antioxidant capacity (%TAC), which refers to the reduced glutathione salivary concentration (GSH). Moreover, the protein concentration and the gelatinolytic activity of metalloproteinases were quantified. Results: The data showed that the pH value is slightly more alkaline in OSAS subjects; however, it is not directly related to the severity of OSAS. The %TAC was found to be significantly reduced by 86.2% in the OG. Proteins of saliva from the OG were found 1.5 times more concentrated than in the healthy sample. The gelatinolytic activity of metalloproteinases of healthy and OSA did not show statistically significant changes. Conclusions: The salivary samples from OSAS compared to CG show an alteration of the oxidative state, the pH buffering power, and protein concentrations, inducing conditions that can easily evolve chronic gingivitis. Further investigations are necessary to evaluate the feasibility of using salivary fluid for the early diagnosis of oral or systemic problems in OSAS subjects.

## 1. Introduction

Obstructive sleep apnea syndrome (OSAS) is a frequent sleep disorder, featured by upper airway collapse during sleep, resulting in intermittent hypoxia and sleep fragmentation. The prevalence rate of OSA is reported as 59.7% among adults [1], children, and adolescents; prevalence rates range between 1.2% and 5.7% [2]. Polysomnography (PSG) remains the gold standard test for diagnosing sleep-disordered breathing. The apnea-hypopnea index (AHI) is used to diagnose OSAS, and it is calculated as the number of apneas and hypopneas per hour during sleep. An AHI score of >5 with clinical symptoms, such as witnessed apneas, excessive daytime sleepiness, loud snoring, and nocturnal choking or AHI > 15 without accompanying clinical symptoms, is diagnosed as OSAS. An AHI score < 5 is accepted within normal limits; and scores between 5–15, 15–30, and >30 indicate mild, moderate, and severe OSAS, respectively. OSAS has been associated with several systemic disorders, such as hypertension, diabetes, and dyslipidemia. Accordingly, large evidence supports the relation between OSAS and cardiovascular risk [3,4,5]. Several mechanisms linking OSAS to its consequences have been identified, and oxidative stress represents one of the main triggering events leading to the increased cardiovascular risk. Moreover, events in the upper airways caused by snoring, hypoxia, nasal congestion, or muscular stress can concur with the systemic oxidative stress. The oxidative stress is defined as the imbalance between reactive oxygen species (ROS) production and antioxidant defense inside the human organism. Accordingly, hypoxia, caused by OSAS, can be accompanied by a disturbance in gas exchange and oxygen desaturation, which may hardly be counterbalanced by the physiological antioxidant capacity of body fluids. Thus, the excessive production of ROS tilts the oxidative balance generating damage to one or more biomolecules, including DNA, RNA, proteins, and lipids. Saliva, as a representative sample of the oral ecosystem, has become an important sample matrix in bioanalytic and it reflects systemic conditions [6]. Salivary composition is associated with oral and systemic diseases and mediating inflammatory responses [7]. The altered composition of saliva with increased concentration of inflammatory biomarkers (e.g., inflammatory cells, interleukin IL-8, and IL-6) has been documented in patients with severe OSAS [8]. In addition, proinflammatory cytokines were elevated in the saliva and serum of OSAS patients [9,10]. Patients with OSAS present an imbalance between the production of oxidizing agents and the activity of the antioxidant counteracting system, showing increased oxidative damage of proteins and compromised antioxidant defenses [11]. Total antioxidant capacity (TAC), defined as the moles of oxidants neutralized by one liter of solution, is a biomarker measuring the antioxidant potential of body fluids [12]. However, to our knowledge, in the literature, the pH determination in the saliva of OSAS patients has not yet been investigated. The evaluation of salivary composition may be a practical, safe, easy, cheap, non-invasive, and rapid method that would be especially suitable for children. It is known that the overall ability of salivary samples to reduce a redox chromogen can be measured by a quantitative biochemical method and normalized by the concentration of salivary proteins [13]. Thus, the aim of this study was to quantify the variation of saliva, protein concentration, and antioxidant capacity in adult subjects with OSAS (OG) compared with a control group of healthy subjects (CG).

## 2. Materials and Methods

From December 2020 until March 2021, a group (OG) of 50 subjects aged between 30 and 75 years (mean age 57.9 ± 11.1 years) was recruited from the Department of Orthodontics and Neurology Unit of the University Hospital of Rome Tor Vergata. All the subjects of the OG underwent nocturnal polygraphic cardiorespiratory monitoring, which was diagnosed according to the American Academy of Sleep Medicine (AASM) criteria [14]. The inclusion criteria of the OG were: adult subjects older than 18 years of age, diagnosis of OSAS confirmed with PSG (AHI > 5 events/h), absence of systematic disease, absence of chronic periodontitis, history of chronic airway disease, cerebrovascular disease, chronic cardiac failure, malignancies, endocrine disease, and mental disorder (depression and anxiety). These subjects were compared with a control group (CG) composed by 20 subjects (mean age 28.4 ± 5.2) recruited from the healthy population. The control group was selected through a careful medical history (that excludes the presence of absence of systematic disease, absence of chronic periodontitis, history of chronic airway disease, cerebrovascular disease, chronic cardiac failure, malignancies, endocrine disease, and mental disorder), ESS test < 10, absence of signs and symptoms of OSAS, and physical characteristics (advanced age, BMI, excessive neck and abdominal circumference, craniofacial characteristics that predispose to OSAS). Therefore, the CG did not perform any polysomnography. The study project was approved by the Ethical Committee of the University of Rome Tor Vergata (protocol number 14/21). A saliva sample was harvested and collected in a test tube from all the subjects of both groups (OG and CG) between 9:00 and 11:00 in the morning, to minimize sampling biases. All the patients did not eat for at least 90 min and rinsed with water before salivary sampling, for washing the oral cavity from gross impurities. A sterile and disposable syringe was used to aspirate the fluid from the bottom of the oral cavity, taking care to collect at least 2 mL of saliva for each subject [15]. The fluid was promptly thawed at a temperature of −20°C and kept frozen until then, analyzed at the Biochemistry Laboratory of Tor Vergata University.

### 2.1. Laboratory Analysis

Samples were centrifuged for 15 min at 4000 rpm at 4 °C and filtered with a 22 μm filter before being analyzed. The precise measurement of pH (accurate to the second decimal place) was carried out using a glass microelectrode as a reference electrode, which is systematically calibrated before carrying out the potentiometric measurement of the sample under examination. The measurement of the salivary protein concentration was carried out by means of the Bradford Assay, evaluating the salivary protein concentration (inversely proportional to salivation), and it was employed for the normalization of TAC samples. The appropriately diluted samples were stained with Bradford blue dye and taking as a reference a standard curve of albumin standard solutions; the absorbance was measured at a wavelength of 595 nm. Using a calibration line, the protein concentrations of each sample were derived from the calibration curve.

Total antioxidant capacity (TAC) is the measure of the amount of free radicals scavenged by a test solution (0.08 mM 2,6-dichlorophenol indophenol, 0.3% *v/v* THESIT detergent, 5 mM Na_2_EDTA 75 mM phosphate buffer pH 8.8) [13]. Briefly, the sample and redox colorant reagent were mixed in the ratio of 1:10, and absorbance was recorded at 595 nm. Reduced glutathione solutions were used for the calibration. Absorbance readings were recorded at t_0_, right after mixing (ABS_t0_); and at t_1_ after 30 min from mixing the sample or calibrator with the redox colorant reagent (ABS_t1_). Total antioxidant capacity is expressed as a percentage and compared to the mmol/L of reduced glutathione by a standard curve calibration according to the following formula: (ABS_t0_ − ABS_t1_)_sample_/(ABS_t0_ − ABS_t1_)_GSH_.

Total antioxidant capacity is expressed in terms of mmol/L of reduced glutathione.

Gelatin substrate zymography was used for the evaluation of saliva MMP gelatinolytic activities in the salivary samples [16]. Briefly, 1.8 μg of saliva proteome were loaded in each lane. The latent forms of metalloproteinases were activated by incubation for one hour at 37 °C with solubilization buffer (0.25 M Tris, 0.8% sodium dodecyl sulfate (SDS), 10% glycerol, and 0.05% bromophenol blue) and then, run into a 12% SDS-PAGE gel containing 1 mg/mL of gelatin type B (Sigma Chemical Co., St. Louis, MO, USA). After electrophoresis, the gels were washed twice with a detergent buffer (2% Triton-X100 in ultrapure water) in order to remove the SDS; subsequently, the gels are incubated at 37 °C for 22 h with an activity buffer (50 mmol/L Tris-HCl, pH 7.5, 10 mmol/L CaCl2, 150 mmol/L NaCl); then, stained with Coomassie Blue R 250 (Bio-136 Rad, Hercules, CA, USA) for 1 h; and then, de-stained to visualize the gelatinase activity bands, clear on a blue background. The intensities of the gelatinolytic activity areas were measured through an image analysis program (IMAGE J free scientific image processing software) and quantified using a scale of arbitrary units (AU). The statistical analysis of data was performed by Prism Software. The correlation test and Student’s *t*-test were used to analyze the statistically significant differences. Data were expressed as mean ± SD.

### 2.2. Statistical Analysis

All statistical analyses were performed with the aid of the statistical software SPSS (Statistical Package for Social Sciences, version16.0, SPSS Inc., Chicago, IL, USA). Descriptive statistics were used to describe both sample groups (OG and CG) in terms of age, gender, and physical characteristics. To assess the normality of the data, the Shapiro–Wilk test was used. The *t*-test was applied to evaluate the comparison of quantitative variables between the groups. Values of *p* < 0.05 were considered significant.

## 3. Results

The OSAS group was composed of 50 subjects (39 M, 11 F) with a mean age of 57.9 ± 11.1 years of age. Moreover, the control group was composed of 20 subjects (6 M, 14 F) with a mean age of 28.4 ± 5.2 years of age recruited from the healthy population. The OG showed a mean AHI/h of 38.8 ± 23.7 obtained by nocturnal PSG. The CG did not perform any polysomnography and there were no data regarding the AHI/h index, which is assumed to be below the pathological threshold. The body mass index (BMI) of the OG was 29.3 ± 5.6, while for the CG it was found to be 21.8 ± 2.1. Table 1 summarizes the clinical and demographic characteristics of the analyzed samples, whereas the biochemical parameters detected on the salivary samples are reported in Table 2. The analysis of the data showed that the pH value is slightly more alkaline in the subjects with OSAS. According to the obtained results, the increasing of the AHI index was not directly related to saliva variation. Furthermore, the statistical analysis did not show a statistically significant difference between the pH values of the two groups (*p* = 0.48). An increase in salivary protein concentrations was found 1.5 times more concentrated in OSAS patients with respect to healthy concentration levels; these variations are directly correlated to the salivary density. Comparing the concentrations of reduced glutathione (GSH), expressed in μM, a lower presence of GSH was found in the OSAS subjects, compared to the CG. This variation between the two groups was found to be statistically significant (*p* < 0.005). The antioxidant power of saliva, expressed as a percentage (%TAC), giving healthy patients a reference value of 100%, was found to be reduced by 86.22% in the OSAS group and this variation was evaluated as statistically significant with a *p*-value < 0.05. Gelatin substrate zymography densitometric analysis evaluated the MMP activities of MMP2 and MMP9 pro-forms found in the salivary samples. Going into detail, the analysis reported showed no significant variation or correlation to the severity of OSAS disease (Figure 1).

## 4. Discussion

The present study focused on the possible variation of salivary pH, protein concentration, and salivary antioxidant capacity in subjects with OSAS compared with a control group. To our knowledge, no previous studies analyzed the correlations regarding changes in pH and antioxidant capacity of saliva in patients with obstructive sleep apnea. The difference between the mean age of the two groups and the non-uniform distribution of genders and ages may represent a limitation of this study. However, a young group of healthy volunteers is more likely to have physiological polysomnography; in addition, the younger age is associated with a minor risk in developing OSAS. The collection method was chosen based on the two different methods proposed by Kubala et al. (2018) [17] to collect saliva: the first is the free flow of saliva not stimulated by the mouth, spitting with little muscle stimulation; and the second consists in aspirating the saliva from the bottom of the oral cavity with sterile syringes. For this study, the second collection method was chosen, as it was considered the method that minimizes contaminating saliva and sampling errors. In the present study, it was also chosen to perform the sampling without salivary stimulation as proposed by Varga (2012), as this way salivary fluid is considered isotonic with plasma. In the same study, Varga states that the pH value of saliva is not a constant; however, it undergoes significant changes under the influence of various factors (such as salivary secretion rate, diet, systemic diseases, and a vegetative nervous system). It has been shown that salivary pH increases significantly after meals [18]. In our study, all the patients did not eat for at least 90 min and rinsed their mouths with water before salivary sampling in the time slot between 9.00 and 11.00. The results of our study did not reveal a statistically significant difference between the pH of the subjects with OSAS (mean pH 7.29 ± 0.57) and the healthy population (mean pH 7.22 ± 0.84). There are no studies in the literature that evaluate this parameter; however, we can compare our results with the results of Baliga et al. [15], who reported that the mean pH of the population with clinically healthy gingiva is 7.06 ± 0.04. The mean pH of the group with chronic generalized gingivitis is 7.24 ± 0.10, while the mean pH of those with chronic generalized periodontitis is 6.85 ± 0.11. Salivary pH is influenced by numerous factors determining its variation during the day; however, according to our results, the increase in the AHI index does not significantly affect the pH of saliva in an OSAS patient. The present study showed a statistically significant reduction in the salivary concentration values of reduced glutathione ([GSH] μM) in the OSAS subject group compared with the control group. Accordingly to our results, Cofta et al., in a study conducted in 2019 on Caucasians with mild to severe OSAS, showed a proportional decrease in the total antioxidant status of plasma to the severity of OSAS [19]. Conversely, plasma and lymphocyte markers analysis of oxidative stress were found higher in OSAS patients with respect to untreated ones [11,18]. Our study showed significantly decreased %TAC levels in the group of subjects affected by OSAS compared to the group made up of a healthy population. Compared with a control group, subjects affected by OSAS show salivary indicators that suggest an alteration of cellular oxidative stress processes: OSAS subjects present statistically significant decreased salivary %TAC levels compared with CG. In conclusion, saliva reacts as other tissue and body fluids to the oxidative stress caused by the continuous cycles of hypoxia and reoxygenation caused by OSAS.

## 5. Conclusions

The focus of this research was to evaluate the variation of salivary biochemical parameters (such as pH, protein concentration, and antioxidant capacity) in subjects with OSAS compared with a control group. According to the results of our study, there is not a statistically significant difference between the pH and the gelatinolytic activity of the salivary samples from OSAS and the healthy population. In addition, the gelatinolytic activity of metalloproteases MMP-2 and MMP-9, which represents an important parameter for the biochemical characterization of saliva, showed no differences both in the OG and CG. Salivary pH is influenced by numerous factors determining its variation during the day; however, the presence of OSAS does not significantly affect the electrolyte composition of saliva. Compared with a control group, the subjects affected by OSAS show salivary indicators that suggest an alteration of cellular oxidative stress processes: OSAS subjects present a statistically significant reduction in salivary concentration values of reduced glutathione ([GSH] μM) and decreased salivary %TAC levels compared with CG. The assessment of the salivary composition can be exploited for the evaluation of oxidative markers and for the early diagnosis of OSAS. Early diagnosis can reduce the risk of related comorbidities. Saliva can be an important practical, safe, easy, non-invasive, and rapid method for early diagnosis. The limitations of our study are represented by the difference between the average age of the two groups and the non-uniform distribution between males and females. Considering the limitations, further investigations on a larger sample of OSAS subjects may allow for new insight into the relationship between salivary antioxidant capacity and OSAS.

## Figures and Tables

**Figure 1 ijerph-19-13219-f001:**
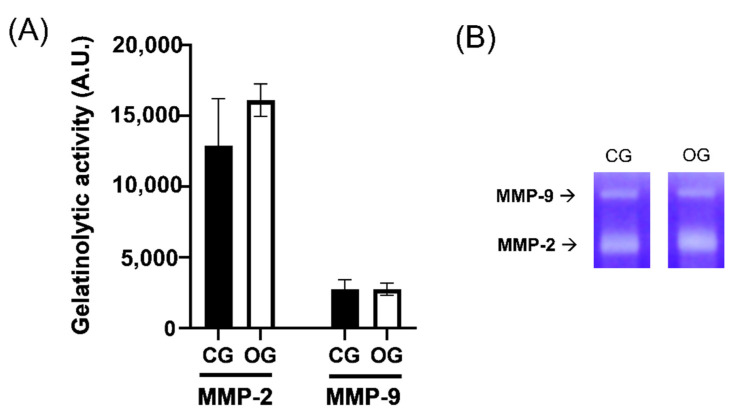
**Gelatinolytic activity of MMPs from salivary samples.** (**A**) Bar plots of the densitometric analysis of gelatinolytic bands from MMP9 and MMP2 activity. (**B**) A representative zymogram: of the gelatinolytic activity of MMP2, and MMP9 pro-forms of MMP gelatinases of salivary samples from control and OSA group (CG and OG, respectively). The intensities of the gelatinolytic activity areas were measured through IMAGE J free scientific image processing software and quantified using a scale of arbitrary units (AU). Statistical analysis was performed by GraphPad MMP-9 (*p* = 0.97) and MMP-2 activity (*p* = 0.19).

**Table 1 ijerph-19-13219-t001:** Characteristics of OSAS group and control group.

	Sex	Age	Mean AHI/h	Mean W (Kg)	Mean H (cm)	Mean BMI
**OG**	39 M; 11 F	57.9 ± 11.1	38.8 ± 23.7	87.3 ± 19.3	172.3 ± 8.8	29.3 ± 5.6
**CG**	6 M; 14 F	28.4 ± 5.2	**/**	63.9 ± 12.0	169.4 ± 9.4	21.8 ± 2.1

**Table 2 ijerph-19-13219-t002:** Mean value of the biochemical analysis.

	Mean pH	Mean %TAC	Mean [GSH] μM	Mean [Protein] mg/mL
**OSAS Group**	7.29 ± 0.57	13.78 ± 7.5	0.06 ± 0.04	3.08 ± 0.08
**Control Group**	7.22 ± 0.84	100	0.51 ± 0.11	2.05 ± 0.11
** *p* ** **-Value**	0.78 (ns)	0.003 *	0.015 *	0.002 *

* *p* < 0.05.

## Data Availability

Not applicable.

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
