# Peer review of "Saliva Analysis of pH and Antioxidant Capacity in Adult Obstructive Sleep Apnea Patients"

_ijerph, 2022, doi:10.3390/ijerph192013219_

Round 1
Reviewer 1 Report (Previous Reviewer 1)
Dear Editor, the authors have tried hard to answer all questions. Due to limitations, the article should be considered as a clinical trial for future research.
I consider the article suitable for publication after minor corrections and clarifications:
a) In the zymography methodology, the use of only 1.8 ug of protein for each sample was described. I suggest you confirm this information, as the amount is very small;
b) Mention the mark of the molecular weight marker used, as soon as it is used as references for the identification of MMP-2 and MMP-9;
c) Tables 1 and 2 need to be corrected. I believe that commas should be replaced by periods in numeric expressions.
Author Response
Dear Editor
thank you to you and to the reviewers for the important suggestions that made our manuscript significantly improved.
We did all the revisions requested by the three reviewers. The revisions are clearly highlighted in the file in red.
The manuscript was corrected in all sections by an extensive English editing, and it was revised in language and form.
REVIEWER 1
I consider the article suitable for publication after minor corrections and clarifications:
a) In the zymography methodology, the use of only 1.8 ug of protein for each sample was described. I suggest you confirm this information, as the amount is very small;
We confirm this was the amount loaded per lane
b) Mention the mark of the molecular weight marker used, as soon as it is used as references for the identification of MMP-2 and MMP-9;
Recombinant enzyme proenzymeMMP2 and MMP9 RD system catalog #902-mp-010 #911-mp-010
c) Tables 1 and 2 need to be corrected. I believe that commas should be replaced by periods in numeric expressions
The tables has been modified as suggested by the referee.
Please see the attachment

Reviewer 2 Report (Previous Reviewer 2)
Abstract
1. Add a sentence about the background of the study in addition to the aim
2. Preferable to use the “period” (.) as a decimal separator for numerical values and not “comma” (,)
Introduction
3. Page 2, lines 70-75 – Mentions a portion of the methodology. Please rephrase it. Also include the aim of the study at the end of the introduction.
Materials and methods
4. Preferable to use the “period” (.) as a decimal separator for numerical values and not “comma” (,)
5. Include the age range of the subjects, along with the mean age
6. Page 2, lines 87-88 – It has been mentioned that the control group subjects did not undergo any polysomnography, while ideally they should have been subjected to PSG. So, how were they ascertained as not having OSA. Was it based on history or any other morphological/clinical characteristics? Without which they can’t be considered the control group.
Results and Discussion
7. There seems to be a significant difference between the mean ages of subjects in OSA group and control group, with the OSA group subjects considerably older. Since it is reported that Salivary antioxidant capacity decreases with increasing age, why not the reduced TAC% observed in the OSA group be the result of higher age and not OSAS?
8. There is nothing discussed about the above issue in the discussion too. In the absence homogeneity of subject ages in both groups, authors need to substantiate the confounding effect of senility on the antioxidant capacity observed in OSA group. Without which the results of the present study hold no remarkable value.
Author Response
RESPONSE TO THE EDITOR
Dear Editor
thank you to you and to the reviewers for the important suggestions that made our manuscript significantly improved.
We did all the revisions requested by the three reviewers. The revisions are clearly highlighted in the file in red.
The manuscript was corrected in all sections by an extensive English editing, and it was revised in language and form.
Abstract
1. Add a sentence about the background of the study in addition to the aim
The text has been modified as suggested by the referee.
2. Preferable to use the “period” (.) as a decimal separator for numerical values and not “comma” (,)
The text has been modified as suggested by the referee.
Introduction
3. Page 2, lines 70-75 – Mentions a portion of the methodology. Please rephrase it. Also include the aim of the study at the end of the introduction.
The text has been modified as suggested by the referee.
Materials and methods
4. Preferable to use the “period” (.) as a decimal separator for numerical values and not “comma” (,)
The text has been modified as suggested by the referee.
5. Include the age range of the subjects, along with the mean age
The text has been modified as suggested by the referee.
6. Page 2, lines 87-88 – It has been mentioned that the control group subjects did not undergo any polysomnography, while ideally they should have been subjected to PSG. So, how were they ascertained as not having OSA. Was it based on history or any other morphological/clinical characteristics? Without which they can’t be considered the control group.
The control group was selected through a careful medical history which included the ESS test and the absence of signs and symptoms of OSAS and physical characteristics (advanced age, BMI, excessive neck and abdominal circumference, craniofacial characteristics which predispose to OAS). This allowed us to assume that the sample was in good health and to avoid polysomnography, an expensive examination.
Authors explained much better the inclusion criteria for the control group selection.
Results and Discussion
7. There seems to be a significant difference between the mean ages of subjects in OSA group and control group, with the OSA group subjects considerably older. Since it is reported that Salivary antioxidant capacity decreases with increasing age, why not the reduced TAC% observed in the OSA group be the result of higher age and not OSAS?
Probably there was a misunderstanding, we apologize for the mistake, the mean values of [GSH]μM of OG and the control group were swapped in the original manuscript (see Table 2). In the revised version table 2 has been corrected. In any case, the results report a significant difference between the mean ages of subjects in OSA group and control group with a decrease in the total antioxidant capacity for OSA group( see table 2).
8. There is nothing discussed about the above issue in the discussion too. In the absence homogeneity of subject ages in both groups, authors need to substantiate the confounding effect of senility on the antioxidant capacity observed in OSA group. Without which the results of the present study hold no remarkable value.
As reported in the text of the manuscript, we are aware that the limitation of our study is represented by the difference between the average age of the two groups. However, we are confident that the changes we have found are relevant because between the OSA and the control group we found a difference of one order of magnitude.
Please see the attachment

Reviewer 3 Report (Previous Reviewer 3)
The quality of revised manuscript improve a lot. But the limitation about the non-even patients age is still my concern. Actually author could choose young OSAS patients or non-OSAS old people to do the study. It will make this study more reasonable.
Author Response
Dear Editor
thank you to you and to the reviewers for the important suggestions that made our manuscript significantly improved.
We did all the revisions requested by the three reviewers. The revisions are clearly highlighted in the file in red.
The manuscript was corrected in all sections by an extensive English editing, and it was revised in language and form.
The quality of revised manuscript improve a lot. But the limitation about the non-even patients age is still my concern. Actually author could choose young OSAS patients or non-OSAS old people to do the study. It will make this study more reasonable.
We are aware that the limitation of our study is represented by the difference between the average age of the two groups. The study group we selected had an average age of 57.9 years. We tried to select a control group of similar age that could allow us to exclude the use of polysomnography (very expensive test) due to the absence of the following characteristics: systematic disease, absence of chronic periodontitis, history of chronic airway disease, cerebrovascular disease, chronic cardiac failure, malignancies, endocrine disease, and mental disorder), ESS test <10, absence of signs and symptoms of OSAS and physical characteristics (advanced age, BMI, excessive neck and abdominal circumference, craniofacial characteristics which predispose to OSAS). This was not possible and the best possible control group was selected. Similarly, we did not have an OSAS group with an average age of 28.4 years.
Please see the attachment

Round 2
Reviewer 2 Report (Previous Reviewer 2)
Thank you for addressing the review comments
Reviewer 3 Report (Previous Reviewer 3)
Considering it is difficult to find control group with similar age as OSAS group. The revised manuscript is acceptable.
This manuscript is a resubmission of an earlier submission. The following is a list of the peer review reports and author responses from that submission.
Round 1
Reviewer 1 Report
Dear Author, I recognize the merit and effort of the team in conducting the research, however, the manuscript has serious problems in its writing. The terminology of biochemical parameters needs to be reviewed and standardized throughout the manuscript. The methodology of biochemical parameters is confusing. The conclusion needs to be more concise. The manuscript needs to make clear the unpublished points that are being addressed compared to the literature already available. Also, there are typographical errors that need to be fixed.
What is the unpublished information of the manuscripts compared to the following published works: a) Peluso I, Raguzzini A. Salivary and Urinary Total Antioxidant Capacity as Biomarkers of Oxidative Stress in Humans. Patholog Res Int. 2016;2016:5480267; b) Tóthová L, Hodosy J, Mucska I, Celec P. Salivary markers of oxidative stress in patients with obstructive sleep apnea treated with continuous positive airway pressure. Sleep Breath. 2014 Sep;18(3):563-70.
Title: I suggest not using acronym in the title. The scope of the journal is generic and involves various aspects of public health. Therefore, the acronym “OSA” may not be familiar to most readers of the magazine.
Summary:
a) I suggest reviewing the question of defining the antioxidant capacity parameter;
b) The description of the methodology is confusing. The way it is written can lead to the understanding that the analyzes of two parameters were carried out: total antioxidant capacity and reduced GSH. This snippet needs to be written better;
c) The results are also mixed. Need to standardize terminology. The terminology switches between TAC and GSH.
d) I consider it inappropriate to associate the results with biomarkers of gingivitis, since no results were presented on the oral health of the volunteers.
Materials and methods
a) What are the criteria adopted for the sample calculation?
b) Was there any index of analysis of the volunteers' oral health status?
c) Please, I ask you to justify the reason for not having performed the polysomnography in the control group.
d) Dear authors, I ask you to justify the reason for not carrying out the pH assessment immediately after the saliva collection. Saliva pH is determined by several salivary components. There are several processing steps that have been reversed. First the raw saliva was frozen, and only after freezing it was centrifuged and filtered to carry out the biochemical analyses. Is it routine for the laboratory to carry out the analysis in this way? The antioxidant capacity tends not to be stable after thawing the samples. Considering that there are several steps (centrifugation and filtration), this could compromise the biochemical analysis.
e) The methodology of TAC and GSH is confusing. The description of the methodology suggests that two different analyzes were carried out: “The measurement of the salivary protein concentration was carried out by means of the Bradford Assay, evaluating both the salivary protein concentration (inversely proportional to salivation), and the %TAC ( salivary antioxidant power expressed as a percentage) and [GSH] glutathione concentration which indicates the reduced antioxidant power.”
f) In the case of salivary biochemical analyses, I strongly recommend that TAC values ​​be normalized by protein content;
Statistical analysis
a) Which statistical test was used to analyze normality?
Results
a) I consider it very serious that the authors did not present the results of the densitometric analysis of the zymograms. In addition, they should have presented a representative figure of the zymogram containing samples from both groups. The authors could have explored the description of the results. Was it possible to identify MMP-2 and/or MMP-9 in the zymogram?
b) The average age between the groups is very different. Could this not influence the results? I believe that there should be a correlation between age group and gender distribution between the control and apnea groups.
c) How was the TAC percentage calculated?
d) Was the value of the GSH concentration in the control group 0.06 umol/L, which corresponds to 100% TAC? However, the GSH value in the OSAS group was 0.51 umol/L, a much higher value compared to the control group. If the CAT was calculated using a GSH standard curve, how can this reduction be explained in the OSAS group?
Discussion
a) I suggest that the discussion be reformulated. Much information entered in the conclusion is part of the discussion.
Conclusion
a) Is gelatinotic activity not considered a biochemical parameter?
b) The conclusion needs to be more succinct and coherent with the proposed objectives. The conclusion must stick to the results obtained without extrapolations.
Reviewer 2 Report
1. Please use standardized abbreviations in all places throughout the manuscript. For the study groups OG and CG, in some areas it has been mentioned as GO and GC.
2. Why did the authors chose young members as part of the control group? While the mean age of patients in the OSA group was 57.9 ± 11.1 years, the mean age in the control group was 28.4 ± 5.2 years. The mean ages are significantly different between the two groups. Couldn't that have been a confounder to the observed results?
3. Similarly, the BMI of the participants in the OSA group (29.3 ± 5.6) and control group (21.8 ± 2.1) are also significantly different. Yet again this could have been a confounder to the observed results.
4. In the absence of any explicitly mentioned inclusion/exclusion criteria, patient selection to the present study seems to be biased.
Reviewer 3 Report
The difference of average age (basically two generation) and uneven gender distribution between OSA patients and healthy control could lead to bias/wrong results. Also, the mean BMI between two groups is different, looks like most OSA patients they chosen are overweight (29.3 ±5.6), control group have normal weight, which also could cause bias.
The fortunate side is that Erden-Inal et al (2002) found negative correlations between age and GSH levels. In this study, the OSA patient group is elder, but GSH concentration is still high. The author should add more to the discussion section about the correlation between age, gender, weight and saliva TAC, GSH, protein since they already noticed this study has limitations.
Also, the author analyzed the gelatin substrate zymography of saliva samples, but there is no figure to show it.
Minor comment:
1. Line 16, please write the full name of OSAS.
2. Line 38, “a large evidence supports…”, please add citation.
3. Line 45, please write the full name of ROS.
4. Line 68, please write the full name of AHI and add more information about AHI.
5. Line 99, the concentration of 2,6-dichlorophenol indophenol is 0.08 µM or 0.08 mM? The reference uses 0.08 mM.
6. Line 133, please write the full name of BMI.
7. Line 137-138, the author said pH variations are not directly related to the severity of this disease, how author get this conclusion?
8. Line 182, I think the present study showed a statistically significant increase in salivary …, not reduction.
9. Line 200-201, how author get the conclusion that “the presence of OSAS does not significantly affect the electrolyte composition of saliva”. There is no data about it.